# Abundant microchondrules in 162173 Ryugu suggest a turbulent origin for primitive asteroids

Matthew J. Genge [1,2] ✉, Natasha V. Almeida [2], Matthias van Ginneken [3], Lewis Pinault[4], Tobias Salge [2], Penelope J. Wozniakiewicz [2,3], Hajime Yano [5,6] & Steven J. Desch[7]

Chondrules are a characteristic feature of primitive Solar System materials and are common in all primitive meteorites except the CI-chondrites. They are thought to form owing to melting of solid dust aggregates by energetic processing within the solar nebula and thus record fundamental processes within protoplanetary disks. We report the discovery of abundant altered microchondrules (>350 ppm) with modal sizes of 6–8 μm within sample A0180 from C-type asteroid Ryugu. These microchondrules have similar log-normal size and shape distributions to normal-sized chondrules, implying evolution by similar size-sorting. We suggest here formation of microchondrules in an outer Solar System chondrule factory, located in the Jovian pressure-bump, followed by turbulent diffusion and concentration relative to chondrules by intense turbulence. Meridional flows could have also separated microchondrules from chondrules and deliver them sunwards of the pressure bump via Lindblad torque flows. Contrary to conventional wisdom we thus propose that the concentration of fine-grained, unprocessed grains could mean the most primitive asteroids did not have to form at the largest heliocentric distances.

Chondrules are ubiquitous in primitive meteorites. In non-carbonaceous (NC) chondrites such as ordinary and enstatite chondrites, chondrules dominate and have abundances of ~85 vol%[1]. In contrast, in carbonaceous chondrites (CCs) fine-grained matrix is abundant, with chondrules comprising <60 vol% of materials[1]. In general, the closer the composition of the chondrite to solar, the fewer and smaller the chondrules it contains, with the CM2 chondrites containing less than 20 vol% chondrules[2], and the CI chondrites, which have the most primitive compositions, having no mm-sized igneous objects[1,3]. Given that models suggest the formation of NC and CC chondrites as two discrete populations separated by the gap in the nebula generated by the early growth

of Jupiter[4], the size and abundance of chondrules thus broadly decreases with heliocentric distance in the early Solar System from chondrule-rich NCs to chondrule-poor CCs.

Samples of carbonaceous asteroid 162173 Ryugu, returned by Japan Aerospace Exploration Agency's Hayabusa 2 mission[5–8], have mineralogies, compositions and textures closely related to CI chondrites and thus have experienced significant aqueous alteration in the early Solar System[9–12]. Hydration occurred owing to the melting of ice by internal heating due to the decay of short-lived radio-isotopes. Like CI chondrites, no mm-sized chondrules have been reported from Ryugu, even though remote sensing suggested these objects might be

[1]Department of Earth Science and Engineering, Imperial College London, London, UK. [2]Planetary Materials Group, Natural History Museum, London, UK. [3]Centre for Astrophysics and Planetary Science, Department of Physics and Astronomy, University of Kent, Canterbury, Kent, UK. [4]Department of Earth and Planetary Sciences, Birkbeck College, London, UK. [5]Department of Interdisciplinary Space Science, Institute of Space and Astronautical Science (ISAS), Japan Aerospace Exploration Agency (JAXA), Sagamihara, Kanagawa, Japan. [6]Space and Astronautical Science, Graduate Institute for Advanced Studies, SOKENDAI, Sagamihara, Kanagawa, Japan. [7]School of Earth and Space Exploration, Arizona State University, Tempe, AZ, USA. ✉e-mail: m.genge@imperial.ac.uk

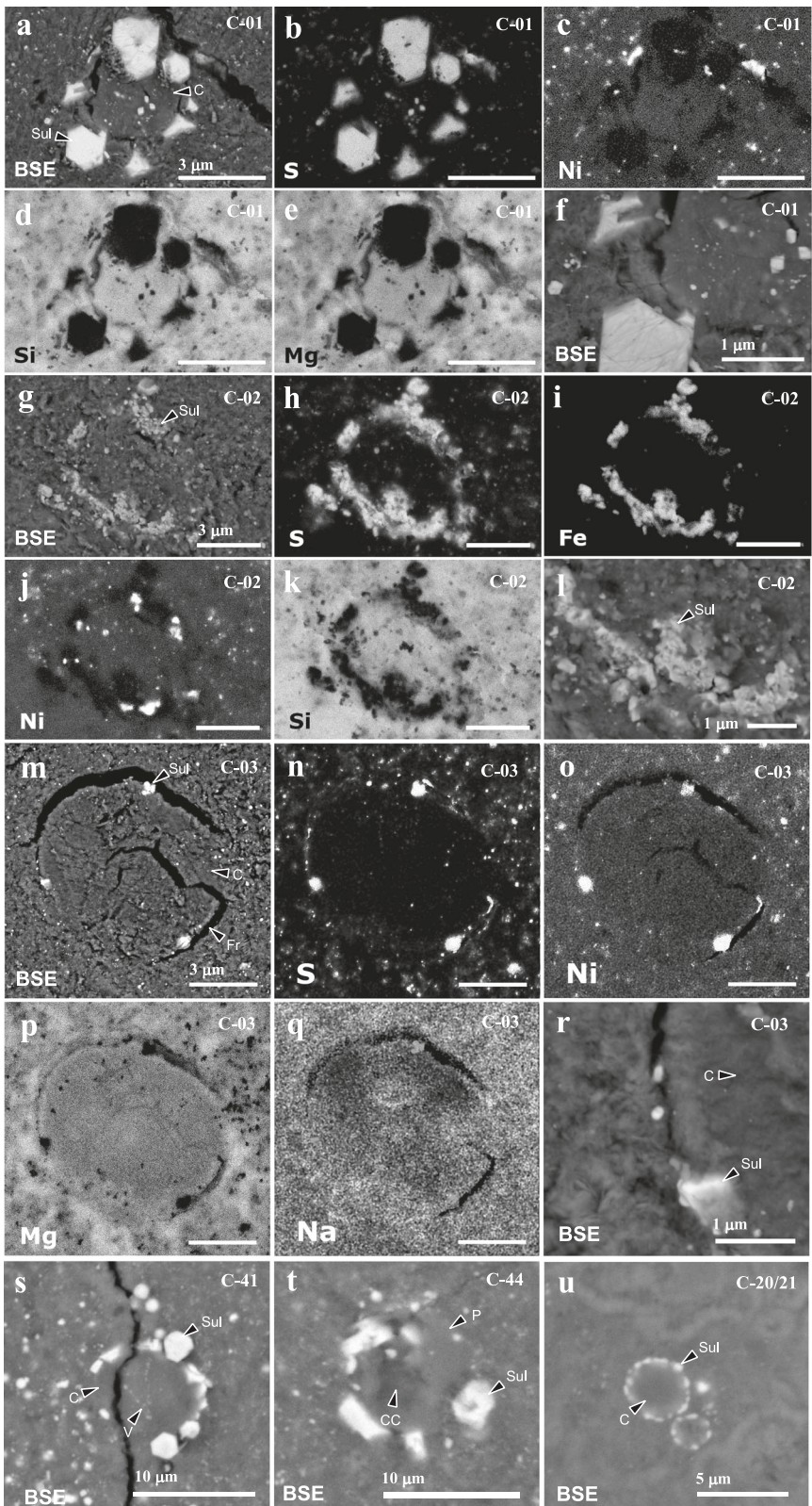

present[8]. Their absence might suggest they were obscured by aqueous alteration, or were not accreted into its progenitor[2].

Small chondrule-like objects 10–20 μm in diameter have been identified in Ryugu samples, but have very low abundances of <20 ppm[13]. These objects are dominated by magnesian olivine similar to Type I chondrules, whilst others have affinities to amoeboid olivine aggregates (AOAs). Their oxygen isotopes lie on the carbonaceous chondrite anhydrous mineral line (CCAM) similar to chondrules from chondrites[14]. The occurrence of similar-sized chondrule fragments in STARDUST samples from comet Wild-2[15] and cluster IDPs[16], thought to be derived from comets, led Nakashima et al.[13], to suggest Ryugu chondrules represent early-formed chondrules that underwent outward radial transport to the formation region of Ryugu in the outer Solar System.

**Fig. 1 | Backscattered electron images (BSE) and element maps of sub-spherical objects (SSO) in Ryugu.** Element maps have been chosen to illustrate the properties of the sulphide rims and the differences between the interior and surrounding matrix. **a** SSO C-01 decorated with euhedral sulphides; Element maps of chondrule C-01 showing (**b**) S, **c** Ni, **d** Si and **e** Mg; **f** High magnification image showing the homogeneous silicate core of C-01, sulphides and surrounding matrix; **g** SSO C-02 showing a sulphide rim surrounding a silicate-dominated core with a texture similar to surrounding matrix; Element maps of C-02 showing (**h**) S, **i** Fe, **j** Ni, **k** Si; **l** A high magnification BSE image of the rim of C-02 showing polycrystalline subhedral sulphide crystals; **m** SSO C-03 showing a sub-spherical core surrounded by fractures decorated with three small Ni-bearing sulphides; Element maps of C-03 showing (**n**) S, **o** Ni, **p** Mg, **q** Na; **r** A high magnification BSE image of C-03 showing the textural difference between the core (right) and the surrounding matrix (left); **s** SSO C-41 showing a homogeneous silicate core with veins of sub-micron iron oxides. The SSO is surrounded by a rim of euhedral sulphide. The object is fractured; **t** SSO C-44 exhibits a core with a facetted anhydrous silicate crystals set in a cryptocrystalline matrix (CC) consisting of phyllosilicate; **u** Two SSOs (C-20/21) with homogeneous silicate cores and sulphide rims. Abbrev: Sul–sulphide, C–core, Fr–fracture, V–vein, P–phenocryst-like phases.

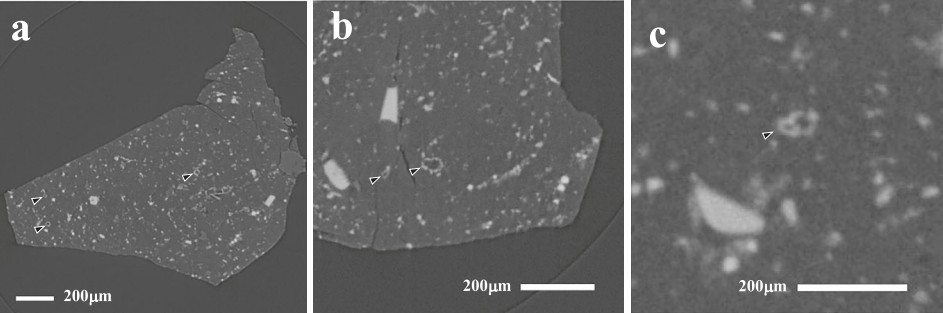

**Fig. 2 | Nano-XCT slices showing sub-spherical objects (SSOs) labelled with arrows. a** Three circular sections through SSOs, all with sulphide rims. **b** Two SSOs, one is elongate and lobate in appearance. **c** A potential compound SSO consisting of two attached parts.

We report the discovery of surprisingly abundant microchondrules in Ryugu sample A0180. All the observed microchondrules are replaced by the products of aqueous alteration and most were recognised by virtue of sub-spherical rims of sulphide in Nano X-ray Computed Tomography (Nano-XCT) data. The occurrence and size distribution of the abundant microchondrules in Ryugu are similar to those of chondrules and suggest size sorting in the early Solar System. We suggest that areas of intense turbulence are required to sort such small particles and will also concentrate primitive components. Turbulent concentration and size sorting of fines of primitive carbonaceous materials may demand high levels of turbulence, higher than expected at 15–30 AU, rather than large heliocentric distance. We suggest that the margins of the gap formed by Jupiter are a possible location for the formation of highly primitive asteroids like Ryugu.

## Results

Sub-spherical objects (SSOs) 5.4–30.2 μm in diameter were observed in sample A0180 and consist of phyllosilicate decorated with a rim of Ni-bearing sulphides (Fig. 1). Sixty one SSOs were observed in a series of polished sections of A0180 (Supplementary Data Fig. 1–3), whilst 108 were observed by Nano-XCT (Figs. 2, 3), where they were identified owing to the contrast provided by their sulphide rims.

In cross-section, all but one of the SSOs are characterised by near-complete rims of iron sulphide crystals surrounding hydrated silicates. The sulphides are too small for accurate quantitative microanalysis (<3 μm); however, elemental mapping indicates they are Ni-bearing, Fe-dominated sulphides similar to the larger pyrrhotite crystals present within the matrix sample (Fig. 1a–f). Most commonly (55% of SSOs) rims consist of tightly interlocking polycrystalline sulphide (Fig. 1g), with all but one of the remainders comprised of separated euhedral sulphide crystals up to 1.5 μm in diameter. Some of these exhibit tabular habits similar to sulphides in the matrix (Fig. 1a). Sulphide crystals can extend outwards into matrix (Fig. 1a, u) or inwards into the silicate core (Supplementary Fig. 1). The sub-spherical shapes of 11 SSOs are abruptly terminated with sulphide rims truncated, suggesting they are fragments of originally sub-spherical objects (Fig. 1s). The final SSO (C-03) lacks a discrete sulphide rim and instead has five sub-spherical Ni-rich sulphide grains located on the periphery of the object (Fig. 1m–r).

The interiors of the SSOs vary in texture and mineralogy. Thirty percent of SSOs have core materials indistinguishable from matrix in texture comprising porous, sub-micron phyllosilicates (Fig. 1g), with three examples containing micron-sized sheet-like crystals of phyllosilicate (Fig. 1t). The majority of SSOs have homogeneous cores of ferromagnesian silicate materials (Fig. 1a, m, s, u), sometimes with sub-micron inclusions of iron oxides (Fig. 1a). The presence of interior or circumference fractures suggests these are hydrated silicates that underwent volume change during hydration. Veins containing sub-micron iron-oxides occur within SSO C-41. One SSO (C-44) has a texture suggesting its core contains porphyritic anhydrous silicates in a mesostasis replaced by porous phyllosilicate (Fig. 1t).

Elemental mapping of SSOs C-01, C-02 and C-03 indicates their silicate cores have more homogeneous Mg and Si contents than the surrounding fine-grained matrix, whilst the interior of C-03 has areas depleted in Na compared to matrix (Fig. 1). Where large enough to analyse, the cores are broadly chondritic in composition (Supplementary Table 1).

The SSOs in A0180 can be identified in Nano-XCT data by the high X-ray attenuation of their sulphide rims compared to the surrounding matrix and their silicate interiors. The SSOs with sulphide rims have concentric structures in XCT slices allowing them to be segmented. Reconstructed 3D models of the objects confirms that sulphide rims form near completed concentric shells (Fig. 3b). Two further examples where sulphides form a hemisphere, were also identified. One SSO also features two connected lobes, suggesting it has a compound structure (Fig. 2c). These three atypical examples were excluded from the size distribution measurements.

The 3D reconstruction of segmented SSOs suggests a modal mean diameter of 6–8 μm with a range from 5.4–30.2 μm (Fig. 4). The size distribution is broadly log-normal, with objects larger than the mode comprising 70% of the population. The SSOs have a small range of sphericity of 0.5–0.67 with a minor decrease with increasing diameter (Supplementary Fig. 4). Irregular step features on the reconstructed SSOs, owing to uncertainty in segmentation, results in some spread to lower values of sphericity, in particular for the smaller objects. The

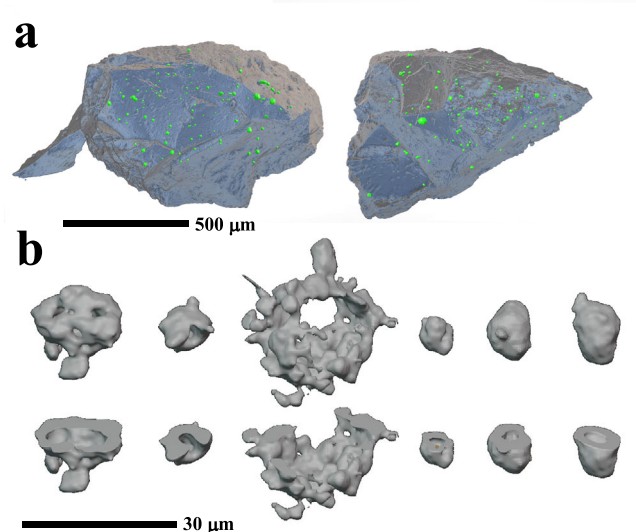

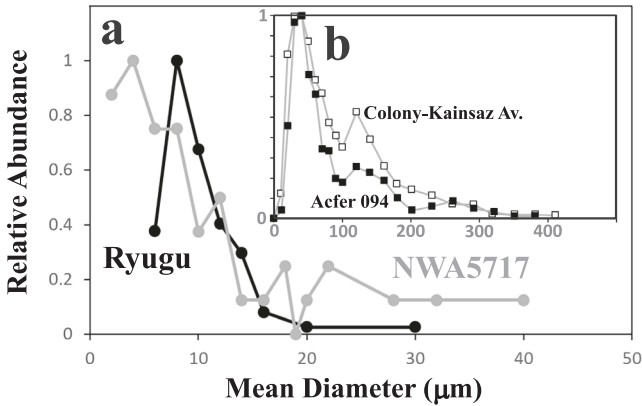

**Fig. 3 | 3D reconstructions of sample A0180. a** Showing abundant sub-spherical objects (SSOs) dispersed throughout the volume of the sample. **b** Showing reconstructions of individual SSOs. The lower images displays truncated structures to illustrate they have a central cavity.

**Fig. 4 | The size distribution of sub-spherical objects (SSOs) in sample A0180. a** Size histogram for SSOs in Ryugu sample A0180 measured from both XCT and in the plane of section and of ungrouped chondrite NWA5717. **b** Apparent size histogram for chondrules in CO chondrites Colony (CO3.0), Kainsaz (CO3.2) and Acfer 094 after Ebel et al.[54].

elongation and flatness of the SSOs indicates most are equant (88%) with nearly equal numbers of prolate (6.5%) and oblate (5.5%) shapes (Supplementary Fig. 4). The total volume fraction of the SSOs determined by XCT is 350 ppm. A total of 61 SSOs, however, were discovered during serial polishing of 7 vol% of the sample with a size range of 2–15 μm in diameter (Fig. 4) and a mode of 6–8 μm similar to the XCT size distribution, even though more smaller SSOs were observed. The abundance of SSOs determined by XCT is thus likely to be a minimum value.

## Discussion

Nakamura et al.[11] reported similar sub-spherical objects to C-03 in Ryugu that they termed phyllosilicate nuggets and were suggested to form by aqueous alteration. The sub-spherical shapes of the sulphide-rimmed objects described here, however, suggests they formed as liquid droplets whose shapes were dictated by surface tension while

molten. The presence of homogeneous phyllosilicate cores within many of the SSOs furthermore implies their interiors comprised anhydrous silicates prior to hydration – most likely glass, owing to the lack of crystal pseudomorphs. Those filled with matrix-like assemblages appear to have been more intensely altered. Fractures within the objects most likely formed by the volume change associated with hydration. Objects with similar textures and compositions have previously been reported in CM2 and hydrated micrometeorites[17] and have been interpreted as altered microchondrules. Sub-spherical phyllosilicate nuggets have also been reported in CI chondrites[3]. In addition to unaltered microchondrules identified in Ryugu by Nakashima et al.[13], Nakamura et al.[12] have identified a single hydrated microchondrule, with a barred texture, completely replaced by phyllosilicates and having a diameter of 30 μm. There is, therefore, excellent evidence that SSOs reported here represent hydrated microchondrules since they are: 1) sub-spherical in shape, 2) have broadly chondritic compositions, 3) are similar to altered microchondrules in CM2 chondrites, and 4) microchondrules with barred textures have been reported in Ryugu. The inferred glassy textures of SSO precursors in contrast are dissimilar to microchondrules reported from Ryugu[12,13], however, glassy microchondrules are common in chondritic meteorites owing to the more rapid cooling of these small droplets[17–20].

A possible alternative explanation for the observed SSOs is formation as a replacement texture during aqueous alteration. Sulphurisation of metal grains during hydration might be expected to generate rims of sulphide surrounding metal grain cores, which could have been subsequently removed by dissolution, allowing the precipitation of phyllosilicates within the produced void space. The high degree of sphericity of some of the SSOs (Fig. 1m, s, u), however, is inconsistent with such a mode of formation. The precipitation of sulphides as drusy linings within cavities might also produce similar textures, indeed an example of such an object was observed in A0180 but has coarser-grained sulphide crystals (>5 μm) that penetrate inwards (Supplementary Fig. 5). These cavity infillings also frequently retain significant void space and contain porous coarse phyllosilicate not observed in SSOs. Nevertheless, two sulphide-rimmed objects were observed in sample A0180 that may have formed by cavity infill and are distinguished from SSOs by their more angular shapes and coarse sulphides (Supplementary Fig. 5). Finally, the presence of hemispheric SSOs, fractures and a tentative composite SSO implies collisional fragmentation and liquid droplet coalescence and thus testifies to a pre-accretional origin for these objects.

The occurrence of microchondrules in sample A0180 differs from those found in chondritic meteorites. In ordinary chondrites (OCs), microchondrules are largely unaltered and are mostly glassy, cryptocrystalline, micro-porphyritic, or radiating pyroxene[17–20]. Two varieties are observed: (1) magnesian microchondrules with compositions similar to the silicate phases in Type I chondrules; and (2) Fe-bearing glassy and vesicular microchondrules[19,20]. Some of these microchondrules have Fe-bearing silicate rims; however, none have discrete rims of iron sulphide or metal. The most likely explanation for the occurrence of sulphide around microchondrules in A0180 is reaction of pre-existing FeNi metal-sulphide rims to sulphides during aqueous alteration. Their concentric geometries strongly suggest they were originally rims on sub-spherical silicate objects, whilst the similarity of euhedral sulphide habits on some of the microchondrules to those in the matrix implies modification by aqueous alteration. Microchondrules with metallic rims have not previously been described; however, armoured chondrules with metallic rims are relatively common, particularly within CR chondrites, and form by the separation of immiscible metal liquid from silicates[1]. Microchondrules in sample A0180, therefore, seem to be distinct from microchondrules observed within ordinary chondrites.

The occurrence of microchondrules in A0180 are also different from those in chondritic meteorites, which are usually concentrated in the fine-grained rims of larger Type I chondrules[18]. Only in the LL3.0 OC Semarkona have abundant microchondrules been found within fine-grained matrix[19]. Overall microchondrule abundances are low in chondritic meteorites, with only sparse (<10 ppm), isolated occurrences described from CM2, CV3 and CO3 chondrites[17,19]. In L and LL OCs, abundances of chondrules <100 μm in size are <60 ppm[21,22]; however, a much higher content of 800 ppm has been reported for the LL3.0 chondrite Semarkona[19]. In this meteorite, clusters of up to 50 microchondrules are reported[19], whilst microchondrules in Ryugu appear to be dispersed in matrix, although A0180, as a millimetre-sized particle, could sample an enriched area.

Dobrica et al.[19,20] presented convincing evidence that microchondrules within OCs are formed as spray from stochastic collisions between chondrules. The concentration of OC microchondrules in the fine-grained rims of Type I chondrules, the existence of protrusions from chondrules spatially associated with microchondrules, and the bulk composition of OC microchondrules are all compatible with their formation as collisional splatter. The absence of whole large chondrules in Ryugu, however, precludes the formation of its microchondrules as a by-product of local chondrule formation, although microchondrules could have subsequently been separated by sorting. Isolated olivine and pyroxene crystals up to 115 μm in diameter, however, are present in CI chondrites and have been suggested to be derived by fragmentation of chondrules on the basis of their oxygen isotope compositions[23]. The lack of distinct igneous textures within such isolated silicates nevertheless argues against this origin.

The microchondrules within Ryugu, therefore, appear to represent a new type of sub-spherical igneous object with affinities to Type I chondrules. Whilst an important new observation, the information that can be derived about their formation processes is limited owing to the degree of aqueous alteration they have experienced. The unaltered microchondrules identified by Nakashima et al.[13] in Ryugu broadly support this conclusion since they are likewise dominated by objects with affinities to Type I chondrules, although none of the three objects observed had sulphide rims. The low abundances observed (<20 ppm) were suggested in part to be the result of destruction by aqueous alteration; however, outwards radial transport from their formation location was also suggested[13].

Given that unaltered microchondrules in Ryugu are rare, it is surprising that sample A0180 contains so many examples. The abundance of >350 ppm is comparable to the highest values seen in chondrule-rich ordinary chondrites. However, only five microchondrules were observed on a single polished surface in sample A0180, with the majority detected by Nano-XCT, thus previous studies may have under-estimated their abundance. The large abundance of microchondrules in A0180 is also likely to be an underestimate since microchondrule C-03 could not be detected in the nano-XCT data owing to its lack of an extensive sulphide rim. Whilst alteration has almost certainly transformed the microchondrules, those with sulphide rims are readily identifiable. The log-normal sizes of the microchondrules is, therefore, specifically for this subset of objects. Sample A0180 lacks anhydrous silicates and is more altered than many Ryugu samples, however, it retains evidence for freeze-thaw processes and may have experienced low-temperature alteration[24].

Notwithstanding the uncertainty caused by aqueous alteration, the size distribution of microchondrules is significant since it is similar to log-normal size distributions of chondrules in chondritic meteorites (Fig. 4), suggesting they experienced aerodynamic size sorting processes[25]. Although it is likely that several different processes can sort chondrules, their size distributions have previously been quantitatively explained by size sorting in nebula turbulence, with the smallest eddies responsible for the concentration of particles within narrow size fractions[25–29]. The smallest eddies in a turbulent system are

defined by the Kolmogorov scale, at which kinetic energy is dissipated by viscosity into heat, and are those with the highest vorticity. Particles with stopping times equivalent to the overturn time of the Kolmogorov scale eddy––i.e., those with Stokes numbers St ~ 1 relative to the Kolmogorov scale or St ~10⁻⁴ relative to the largest-scale eddies––are preferentially concentrated by turbulence[25,26]. If particle density became sufficiently high in turbulent regions in the solar nebula it would have triggered streaming instabilities through back reaction by slurries of gas and dust decoupling from the surrounding gas, resulting in rapid accretion of planetesimals that would preferentially sample the concentrated particles[28]. Recently[30] argued for a different cascade model that predicts different sizes of particles concentrated in different sizes of eddies, and questioned whether particles with $St < 10^{-2}$ could be concentrated at all, due to disruption of clumps by ram pressure. Given that the size of chondrule concentrated in asteroids appears universally uniform, and given the success of disk models predicting the size of chondrules that would be concentrated[28], we will continue to use the findings of ref. 26.

Desch et al.[28] provide the following expression based on Cuzzi et al.[26] that relates the size $d$ of particles concentrated by turbulence, the gas surface density $\Sigma$, and the turbulent parameter $\alpha$––a measurement of the intensity of turbulence.

$$d(\mu m) = 560(T/100\,K)^{0.25}(\Sigma/1000\,g\,cm^{-2})^{0.5}(\alpha/0.0001)^{-0.5} \quad (1)$$

This expression was used by Desch et al.[28] in a one-dimensional model of the Solar Nebula to successfully replicate the sizes of chondrules observed in chondritic meteorites of different ages and heliocentric distances of formation. Equation 1 is based on the assumption that the clustering scale is the Kolmogorov scale and is thus a special limiting scenario in isotropic turbulence that leads to concentration of chondrule-sized objects in the inner Solar System. As mentioned above, direct numerical simulations of particle clustering in turbulence in contrast suggests clustering of larger particles in turbulence under the most likely conditions for planetesimal formation[30].

Desch et al.[28] suggested the CI chondrites formed at ~15 AU at ~3 Myr, with $\Sigma = 6.2\,g\,cm^{-2}$, $T = 46\,K$, and $\alpha = 10^{-5}$, yielding a value of $d = 115\,\mu m$, similar to the isolated anhydrous silicates seen in CI chondrites. However, if Ryugu formed at similar or greater heliocentric distance then unreasonably large values of α would be required to concentrate microchondrules; for example, at 30 AU $\Sigma \approx 10\,g\,cm^{-2}$, $T \approx 40\,K$, and $d = 7\,\mu m$ would require $\alpha = 4 \times 10^{-3}$. This is problematic, because such high α would quickly (<10⁵ yr) mix the NC and CC reservoirs. ALMA measurements also generally favour $\alpha \ll 10^{-3}$ in disks[29], whilst Desch et al.[28] estimates values in the inner Solar System (<1 au) of $\alpha = 5 \times 10^{-4}$ with the outer Solar System (>10 AU) having $\alpha < 10^{-5}$ with α falling as a power law in between. This seems to suggest that microchondrules in Ryugu would not be concentrated at this size by turbulence outwards of the Jovian pressure bump.

The formula above applies even if microchondrules are swept up into larger aggregates, as the aerodynamic stopping time of fractal aggregates is not much different from the stopping time of the smallest constituents. The unusually high abundance of microchondrules in Ryugu compared to other carbonaceous chondrites thus demands a distinct formation environment.

Conditions in the pressure bump just outside the gap opened by Jupiter in the disk are unique and may be the right environment to aerodynamically concentrate microchondrules. A salient feature of the pressure bump is that large (>mm- to cm-sized) particles are concentrated at its center (Fig. 5). Throughout the nebula, pressure tends to decrease monotonically with increasing distance from the Sun. Because the gas is partially supported against gravity by an outward pressure force, it has lower orbital speed, and particles experience a headwind. The resultant drag force removes their angular momentum and they radially drift toward the Sun[25]. However, after Jupiter opened

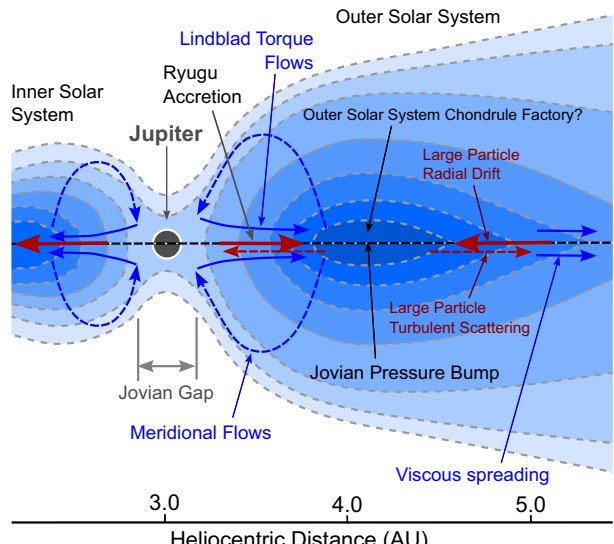

**Fig. 5 | A schematic process diagram illustrating the astrophysical environment that led to microchondrule enrichment in Ryugu.** Blue arrows show gas flow, red arrows show migration of large dust particles. Dashed lines represent weaker flows. Two scenarios are possible: (1) the outward transport of microchondrules coupled to gas through viscous spreading or turbulent diffusion, and (2) the inwards transport of microchondrules by cycling through meridional and Lindblad torque flows. Concentration of microchondrules is most likely during inwards transport and could be enhanced by concentration of small particles within highly turbulent, low density regions near the Jovian gap. Distances are based on the models of Bodenan et al.[38]. The figure is based on observations of a proto-planetary disk by Teague et al.[47].

a gap in the nebula (by ~0.7 Myr after the formation of CAIs[28]), the gas pressure immediately outside the gap increased outwards, reaching a maximum about 1 AU past Jupiter, 4 AU from the Sun[28]. Within the inner half of the pressure bump, large particles drifted outwards, whilst those at larger heliocentric distances drifted inwards towards the center of the bump, resulting in a dust trap that concentrated mm-to cm-sized pebbles[28]. The drag of particles on gas also results in back reaction that can influence gas motion[29]. It is in this region, beyond Jupiter, that chondrule-rich carbonaceous chondrites are likely to have accreted through the concentration of solid particles to sufficiently high values to trigger streaming instabilities (particle mass density equal to gas mass density)[28,29,31–33].

It is very likely that chondrules formed in this same environment. From compound chondrule frequencies, chondrule formation must take place in a region of high density of chondrules (or their precursors)[1]. Chondrule fragments are found in cometary samples[34], suggesting formation beyond Jupiter: the age of individual chondrules ranges from 1.7 to 2.9 Ma after the formation of CAIs[35], and thus postdates the development of the Jovian gap, which represents a barrier to the outwards drift of chondrules (at least through viscous spreading of the nebula). Finally, chondrule-matrix complementarity[36] suggests formation of chondrules in the same region that the chondrites formed. Nebular shocks could have swept through the pressure bump region, forming chondrules, generated by the passage of planetary embryos on eccentric orbits[37]. Some whole-disk models also suggest Jupiter generated spiral density waves with net outwards motion that would have swept through this region[38].

Gas at the inner edge of the pressure bump would have been relatively depleted in larger particles, but not particles the size of microchondrules, which would remain well-coupled to the gas. Shock waves passing through this region would melt whatever precursors were there. At the center of the pressure bump, larger, chondrule-sized

particles would have dominated over microchondrules; at the inner edge of the pressure bump, the few microchondrule-sized precursors may have dominated over larger particles, especially later (several Myr) in disk evolution as larger particles were depleted.

The sunward inner edge of the Jovian pressure bump is an intriguing possible environment for the concentration of small particles. At 3.2 AU and 4 Myr, the model of[28] predicts $\Sigma \approx 8\,\mathrm{g\,cm^{-2}}$, $T = 90\,\mathrm{K}$, which for $\alpha \approx 10^{-2}$ would predict $d = 5\,\mu\mathrm{m}$ using Eq. 1. The Stokes numbers of such particles are $\mathrm{St} = (2\pi)^{1/2}\,\rho_s\,a/\Sigma \approx 2 \times 10^{-4}$. At this location the turbulent wake of Jupiter generates vortices that increase the turbulent coefficient[32,39], to values as high as $\alpha \approx 0.01 - 0.03$ in the vicinity of Jupiter[40]. Turbulence is not only able to concentrate particles of the same size as microchondrules, it might concentrate them to the levels needed to trigger streaming instability. Concentrations of a factor of a few above the background solar nebular solids-to-gas ratio are generally thought necessary to trigger planetesimal growth by streaming instability[41,42], although recent models caution that turbulence can also resist the formation of dense enough regions to cause instability[43]. Turbulent concentration can lead to solids-to-gas ratios enhanced by factors of $10^1$–$10^2$ above the background, in regions of size $\sim\alpha^{1/2}\,H$, where $H$ is the scale height of the disk[26,44]. At $r = 3.5$ AU, assuming $\alpha \approx 0.01$ and $H \sim 0.05\,r$, these regions would be $\sim 3 \times 10^6$ km in size. Each region would contain enough mass of microchondrules (assuming they make up 10% of the solids mass and are concentrated by a factor of $10^2$) to make a planetesimal $\sim 20$ km in diameter−the size of Ryugu−if streaming instability is triggered. There would be >1500 such dense regions created repeatedly over $\sim 10^6$ years at the inner edge of the pressure bump in an annulus 0.1 AU wide, affording many opportunities to trigger the growth of asteroids containing abundant microchondrules.

Conventional wisdom suggests that CI chondrites, as the most primitive and hydrated chondritic meteorites, formed further from the Sun than other chondrites since they accreted less processed materials[10,13]. Previous studies of Ryugu have found higher $\delta^{18}\mathrm{O}$, $\Delta^{17}\mathrm{O}$, $\delta^{15}\mathrm{N}$, $\varepsilon^{54}\mathrm{Cr}$[9,11] than CI chondrites suggesting Ryugu is more primitive, and that the Fe, Ti and Cr isotope systematics of Ryugu and CI chondrites are distinct from other carbonaceous chondrites, but similar to each other[45]. Furthermore, the ratio of $^{16}\mathrm{O}$-rich to $^{16}\mathrm{O}$-poor minerals within Ryugu and CI chondrites is higher than other carbonaceous chondrites and similar to comet 81/P Wild-2[46]. These observations have been interpreted to mean formation of Ryugu at large heliocentric distance, based on the assumption that processing of presolar materials, the carriers of nucleosynthetic signatures, and volatile materials, increases sunwards in the early Solar System. Additionally, $CO_2$ within fluid inclusions inside sulphides in Ryugu have also been used to imply formation beyond the $CO_2$ snow line[11,12]. These properties all seem to favour outwards transport of microchondrules by viscous spreading to large heliocentric distance, far from the chondrule factory, but the large abundance of microchondrules in Ryugu relative to other chondrites is problematic since they would have been diluted by primitive fines during this transport.

An alternative scenario for the formation of Ryugu is the inward cycling and concentration of fine-grained materials in or near the gap formed by Jupiter. The primitive chemical and isotopic nature of CI-like materials and the large abundance of presolar nuclides could be explained by their formation in regions that preferentially trap fine, largely unprocessed, primitive grains, and expel processed larger grains such as chondrules, whilst retaining microchondrules. Thus, CI chondrites might not be fine-grained because they are primitive, but they may be primitive because they are fine-grained. Separation of microchondrules from chondrules could also be facilitated by meridional flows that sweep gas, and coupled fine-dust, to higher disk latitudes and inwards towards the Jovian gap (Fig. 5)[47]. Lindblad torques could then return a portion of this materials outwards at the midplane delivering microchondrule-bearing fine-grained materials. The evidence for the formation of Ryugu beyond the $CO_2$ snow line,

furthermore, could be alternatively explained by exsolution of $CO_2$ gas within carbonated water during aqueous alteration. Indeed, that $CO_2$ is trapped within minerals that were precipitated by water is not consistent with $CO_2$ ice, which would have sublimated earlier in heating, at much lower temperatures than the melting of water ice, and would have been lost. The formation of Ryugu inward of the Jovian pressure bump would predict the presence of rare large chondrules in Ryugu owing to turbulent scattering from the nearby chondrule factory. The high abundance of microchondrules in Ryugu would then be a fingerprint of the formation of CI chondrites in highly turbulent, rather than highly distant regions of the solar nebula.

## Methods
### Sample preparation
Sample A0180 is 1.592 × 0.756 × 0.985 mm in size and was captured in the Hayabusa2 sample container chamber A from the surface of Ryugu (Supplementary Fig. 6). The sample was obtained from JAXA in a sealed container in a nitrogen atmosphere. The initial morphological inspection was conducted on the sample exterior through the container window, with the CLOXS digital optical microscope system on automated digital sample stages developed at JAXA/ISAS[48]. The sample was decanted for Nano-XCT. During the mounting process the sample split into two along fractures (parts A and B). Sample B was mounted for observation.

After nano-XCT analysis the sample was embedded in a Specifix resin and polished under ethanediol with 0.1 μm aluminium oxide. The sample was stored in a desiccator prior to scanning electron microscope (SEM) analysis and in a sealed sample container after analysis. The polished block was carbon-coated prior to SEM analysis. The sample was then serially polished 37 times to expose new planes of section at separations of up to 1 μm. The depth of the polish was determined from the dimensions of spheroidal magnetite by assuming these were spherical and has an uncertainty estimated at 30%. Plucking out of SSOs was the main issue encountered during serial polishing, however, the presence of a cavity from which the SSO was removed can be used to distinguish objects that were hemispherical.

### Nano-XCT analyses
The Ryugu sub-samples (A/B) were mounted in pipette tips and scanned using a Zeiss Versa X-ray micro-computed tomography scanner. X-rays were generated from a tungsten source, with a voltage of 90 kV and a current of 89 μA, using the inbuilt LE4 filter to reduce beam hardening effects. For each scan, 2401 projections were collected across a 360° rotation. Each projection was magnified by a 4x objective lens and recorded using a 2000 × 2000 CCD plane (16-bit pixel depth) at exposure times of 33 s and 28 s for A0180-A and A0180-B respectively, with no binning. Spatial resolutions (in voxels) were 0.625 μm for A-0180A and 0.672 μm for A-0180B. Image stacks were created using the Zeiss Reconstructor software.

**CT segmentation and data processing.** Nano-XCT data were segmented using Image/J and Avizo in order to generate 3D models of specific textural and mineralogical features. The plugin 3DSuite[49] was used to measure sizes from pre-segmented images. The segmentation techniques used varied depending on the feature analysed owing to X-ray attenuation contrast between phases. Initial manual segmentation was performed to mask the exterior of the sample to ensure noise outside the sample was not selected. This step greatly simplified 3D model reconstruction. An initial threshold segmentation was performed on the masked data to determine the volume of the particle, giving a value of 0.77 mm³.

Microchondrules were segmented by virtue of their sulphide rims, which form partial shells of high X-ray attenuation surrounding subrounded regions of low X-ray attenuation consistent with silicates. Segmentation by thresholding was used to generate detailed 3D

models of larger microchondrules as shown in Fig. 4b. These models confirm that sulphides form partial to complete shells surrounding silicate cores. Thresholding, however, could not be used to evaluate the size and shape distribution of microchondrules since this required interpolation in examples with partial rims and inclusion of the central silicate core in the volume. Manual segmentation was used to delimit the exterior of microchondrules for size and shape factor measurements. The uncertainty of manual segmentation was determined by repeat measurement of a cavity in the sample, with ten manual measurements performed and suggests an uncertainty at 2σ of ~10% for diameter (Supplementary Table 2).

### Scanning electron microscopy
Scanning electron microscopy was performed on a polished block using a Hitachi TM4000Pro Desktop SEM at Imperial College London. Backscattered electron images were collected at an accelerating voltage of 15 kV and have resolutions of up to 7 nm per pixel with some variation owing to working distance. Semi-quantitative analyses were obtained by energy dispersive spectrometry (EDS) using an Oxford Instruments AZtec series, silicon drift detector (SDD) with an energy resolution [full width at half maximim (FWGM)] of 151 eV (Cu-Ka). Analyses were performed at an accelerating voltage of 20 kV using a beam current of 2 nA. No gain calibration was applied and thus analytical totals are not quantitative. Oxford instruments matrix corrections were used against factory standards. Uncertainty in elemental ratios was evaluated by analyses of San Carlo olivine and are <5% for major elements (Mg, Al, Fe, Si) and <20% for minor elements (Ca, Mn, Ni). Detection limits are ~0.1 wt% on carbon-coated, polished sections.

Qualitative elemental mapping was carried out at high spatial resolution using a FEI Quanta 650 field emission SEM equipped with a Bruker Quantax EDS system, Esprit 2.5 software, and an annular, four-channel XFlash FlatQUAD SDD (FWHM Mn Kα: 124 eV). The annular SDD is inserted between the pole piece and sample within the chamber of the SEM. The four integrated 15 mm² SDD segments are arranged in radial quadrants around a hole through which the electron beam passes. This geometry allows data collection samples with substantial surface topography. To limit the interaction volume of emitted X-rays and enhance the spatial resolution for elemental analysis to the sub-micrometre scale, a low accelerating voltage of 6 kV has been applied. Hyperspectral imaging datasets that provide complete spectra for each pixel of the SEM image were acquired at a pixel sizes ranging from 13 to 15 nm. The element distribution has been displayed as net intensity elemental maps. Here, the Bremsstrahlung background has been removed and the X-ray line families with overlapping peaks (e.g., Fe-L and Ni-L) were deconvolved using stored line profiles.

Major and minor element compositions were determined using a JEOL 8530 F FE an electron probe microanalyzer (EPMA) equipped with five tuneable wavelength dispersive spectrometers hosted at the Natural History Museum in London. Operating conditions were 40 degrees take-off angle, acceleration voltage of 15 kV, a beam current of 20 nA and focused beam diameter. Elements were acquired using analysing crystals LIFL for Fe kα, Mn Kα, Ni kα, Co kα and Cr kα, PETH for K kα, Ca kα, Ti kα and P kα, and TAPL for Na kα, Al kα, Si kα and Mg kα. The standards were orthoclase for K kα, wollastonite for Ca kα, rutile for Ti kα, Sc-phosphate for P kα, Mn-titanate for Mn kα, fayalite for Fe kα and Si kα, bunsenite for Ni kα, cobalt metal for Co kα, eskoalite for Cr kα, jadeite for Na kα, corundum for Al kα, forsterite for Mg kα. The on-peak counting times were 10 s for K kα and Na kα, 20 s for Ca kα, and 30 s for the rest of element peaks. Sodium and potassium were analysed first to minimise alkali migration during analyses. Background intensities were acquired before and after each peak position, with counting times equal to half of the respective on-peak's acquisition time. Unknown and standard intensities were corrected for deadtime using the normal (traditional single-term) correction method. Interference corrections were applied to Fe for interference

by Mn, Mn for interference by Cr, and Co for interference by Fe. Oxygen was calculated by cation stoichiometry and included in the matrix correction. The matrix correction method was Phi-Rho-Z method, utilising the simplified Puchou-Pichor (XPP) algorithm. Mass absorption coefficients were taken from the FFAST Chantler dataset.

## Data availability

The Hayabusa2 Ryugu sample curatorial dataset is also referenced at https://doi.org/10.17597/ISAS.DARTS/CUR-Ryugu-description[50]. Raw Nano-XCT data is available for download from https://figshare.com/s/eacc0dafe459529093c1 (4 Gb)[51]. An interactive 3D reconstruction of the Nano-XCT data is also available for Windows 64-bit[52] and Mac 64-bit[53]. All other data is provided in the Supplementary Information.

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

## Acknowledgements
The A0180 sample has been loaned to the proposal PI (H.Y.) of this research consortium since 2022, by the JAXA/ISAS Astromaterial Science Research Group through the first international Ryugu sample AO. The authors are the most grateful for the Hayabusa2 project team to have successfully returned the indigenous samples from Ryugu to the Earth in 2020.

## Author contributions
The research was conducted on a sample loaned from JAXA Astromaterials Science Research Group (ASRG) on a proposal for the 1st Announcement of Opportunity for Hayabusa2 samples (AO1) which was led by H.Y., with the other authors as Co-Is. Initial inspection and sample handling were conducted by H.Y. and L.P. Nano-XCT data was collected by N.A., and processed by M.G., M.v.G. and N.A. Electron microscope data was acquired by M.G., M.v.G., P.W. and T.S. All authors contributed data interpretation and science discussion with turbulence calculations by S.D. The manuscript was prepared mainly by MG with contributions from all authors.

## Competing interests
The authors declare no competing interests.
