## [Transparent Peer Review file · Nature Communications]

Abundant microchondrules in 162173 Ryugu suggest a turbulent origin for primitive asteroids

Corresponding Author: Dr Matthew Genge

Version 0:

Reviewer comments:

Reviewer #1

(Remarks to the Author)
See the attached file.

Reviewer #2

(Remarks to the Author)
What are the noteworthy results?

Yes, this study focused on samples returned from a spacecraft mission that sampled asteroid 162173 Ryugu.

- Will the work be of significance to the field and related fields? How does it compare to the established literature? If the work is not original, please provide relevant references.

The work is original, as the first observations and measurements were made from return samples from a primitive asteroid. The discovery of abundant micro-chondrules is interesting for a broad community and understanding of the early Solar System history. The authors make the case for a higher turbulence as traditional models propose.

- Does the work support the conclusions and claims, or is additional evidence needed?

A discussion of the alteration process is needed for the non-specialist. Currently, the manuscript, while clear and concise, does not intrude on the goals of the study or the scientific question or hypothesis to test.

- Are there any flaws in the data analysis, interpretation and conclusions? - Do these prohibit publication or require revision?

I did not find the supplementary materials with the methods; it may have been something I missed, but I was not able to find them in the documents available to me, and without these materials, it is not possible for me to answer this question.

- Is the methodology sound? Does the work meet the expected standards in your field?

My recommendation for the authors is to extract mineral information from the nano-CT, as done recently on Martian meteorites.

The study will benefit from a numerical model that supports Figure 5. It is unclear how this was constructed without a numerical way to test these hypotheses.

- Is there enough detail provided in the methods for the work to be reproduced?

I did not find the supplementary materials with the methods. If this was my flaw, I would request the editor to forward them to me, otherwise the authors must re-submit with complete methods.

Reviewer #3

(Remarks to the Author)
Please see attached document.

Version 1:

Reviewer comments:

Reviewer #1

(Remarks to the Author)
See the attached file.

Reviewer #2

(Remarks to the Author)
The authors made all the requested changes, and I think the paper should be accepted for publication. I have no further suggestions.

Reviewer #3

(Remarks to the Author)
I appreciate the authors' efforts in addressing my concerns, and they have done so for the majority of my suggestions. However, I still have some remaining issues that I would like to ensure they fully grasp. Regarding their revision in response to my comment #5, they added the following sentence on line 218 of their manuscript:

"Equation 1 is based on the assumption that the clustering scale is the Kolmogorov scale and is thus a special limiting scenario in isotropic turbulence that leads to concentration of chondrule-sized objects in the inner Solar System. Direct numerical simulations of particle clustering in turbulence in contrast suggest clustering of larger particles in turbulence [29]."

This statement mischaracterizes the conclusions of Hartlep & Cuzzi (2020). Their work does not claim that larger-sized particles are necessarily the ones preferentially concentrated. The core idea behind the Turbulent Concentration mechanism is that the strongest intermittency in particle accumulation occurs when the turbulent eddy Stokes number St_{η} is approximately 0.3 (as I noted in my previous review). This condition can arise under a broad range of local parameters, including turbulent intensity (α), length-scale of interest (λ), local disk mass (Σ), particle size (d), and other factors that collectively determine $St_{\eta}(\alpha, \lambda, \Sigma, d, \dots)$.

The point emphasized by Hartlep & Cuzzi (2020) (though admittedly, their paper is rather dense in its exposition) is that while Desch et al. (2020) assume a priori that maximal clustering occurs at the Kolmogorov scale (L_{kol}) this is only a subset of the broader conditions that can yield $St_{\eta} = 0.3$. Consequently, the revised sentence's implication that "larger particles" are necessarily concentrated in the inertial range of turbulence is not an inherent outcome of turbulence itself. Instead, the key insight of Hartlep & Cuzzi (2020) is that for a given particle size, d , significant clustering can occur at scales λ that is much larger than L_{kol} . The revised text should clarify this point more accurately -- and simply saying that the HC20 it predicts larger sized particles is incorrect.

The authors response to my comment #5 is insufficient. They can quite easily make an estimate for the more traditional Stokes number, St_K , as defined by the stopping time ($t_s = d^2 \rho_g / (cs^2 \rho_p)$; where ρ_p is the density of the individual solid dust particle, and cs is the local sound speed) multiplied by the local Keplerian frequency (Ω_K). They have the requisite model assumptions already in place, as ρ_g can be derived from Σ (i.e., the same input value range used in their Eq. 1), and cs from T . Providing a range of implied St_K values based on their findings should be a straightforward extension and at least mentioned in a sentence in the main text and then followed by a short exposition in the supplemental materials.

This is crucial because an estimated range of effective Stokes numbers is necessary for particle-gas turbulence modeling. Understanding the implied conditions for clumping based on observational data allows us to better test formation dynamics and compare theoretical predictions with actual data. This is why I ask this.

Response to Reviews

Abundant microchondrules in 162173 Ryugu suggests a turbulent alternative to the distant origins of primitive asteroids

Genge et al.,

General Statement

We have improved the manuscript significantly thanks to the reviewer's comments. We have added a large amount of new data to support our previous, largely as extended data. There have been relatively few changes to the main text. The major changes are:

1) We have performed serial sectioning of the specimen giving 37 new planes of section in which we found 61 SSOs. This is a significant increase from the 5 on which initially presented. These include some lovely examples, which greatly strengthens the evidence that these are microchondrules. In particular there are some great fractured microchondrules that demonstrate these are preaccretional.

2) The newly discovered 61 SSOs were found in 8% volume of the sample, compared to the 108 observed by XCT. This allows us to specify that the 350 ppm identified in XCT is a minimum estimate. We've measured the sizes of SSOs in section and found a similar size distribution to the XCT, indeed the new data includes many smaller SSOs that confirm the log-normal size distribution. The mode of the size distributions is similar.

3) We have included a couple of sentences on objects in A0180 that resemble SSOs but are not consistent with microchondrules (we've called the sulphide-rimmed objects). This allows us give more rigorous criteria for the identification of microchondrules.

3) We have updated the astrophysical aspects of the paper according to reviewer 2's helpful (if very complex) suggestions.

Below we respond to all reviewer comments providing our response and details of changes in italics. For your convenience we provide line numbers for changes. I tried to upload a tracked changes version of the manuscript, however, the PDF conversion stripped out all the annotations.

Reviewer #1 Daisuke Nakashima

Comment 1

The argument that the SSOs are microchondrules is based only on their spherical shapes, which is weak. Since the SSOs and surrounding matrix are both composed of phyllosilicate, so that there is no mineralogical difference between them. In case of chondrule-like objects in the Ryugu samples (Nakashima et al. 2023), they consist of olivine and Fe-Ni metal and have 16Orich and -poor isotope ratios. Even if altered, it shows a barred olivine-like texture, so that it was recognized as a chondrule (Nakamura T. et al. 2022 Science). If the authors observe less-altered clasts in the Ryugu samples, unaltered microchondrules like those (cryptocrystalline?) in ordinary and CM chondrites may be found, though such an object was not found in the clasts in the Ryugu samples as far as I have observed (Nakashima et al. 2023 Hayabusa Symposium).

- The evidence that SSOs are microchondrules are: 1) sub-spherical shapes, 2) silicate-dominated objects with minor sulphide, 3) they have broadly chondritic compositions in major elements, 4) sulphide-rims (similar to type I chondrules in some chondrites), 5) microchondrules have been observed in Ryugu in small numbers indicating their presence. Finally an abstract at MetSoc describes the least altered clast found so far in Ryugu and it includes "chondrule-like" glassy spheres of ~10 μm in size (Matsumoto et al., 2024). I believe it is against Nature Coms policy to reference abstracts (?), however, it should give the reviewers more confidence on our identification of microchondrules. The much larger number of SSO examples we now give are hopefully much more convincing.

Daisuke quite rightly points out that previously identified microchondrules from Ryugu have either porphyritic or barred olivine textures, even if altered, suggesting these were glassy. However, we only can observe microchondrules in XCT if they have sulphide-rims, these rarer porphyritic and barred microchondrules may well be present. In non-carbonaceous chondrites, in which glassy microchondrules are common. Thus homogeneous microchondrules formed by replacement of glass are entirely consistent with microchondrules, indeed those with barred and porphyritic textures are slightly anomalous since such objects should cool more rapidly. Finally, the one example of a microchondrule found in carbonaceous chondrites is an aqueously altered homogeneous sphere - thus there is a precedent for identifying such objects as microchondrules.

To address Daisuke's comments I have added the following:

Line 126 "There is, therefore, excellent evidence that all these sub-spherical silicate objects SSOs reported here represent hydrated microchondrules since they are: 1) sub-spherical in shape, 2) have broadly chondritic compositions, 3) are similar to altered microchondrules in CM2 chondrites, and 4) microchondrules with barred textures have been reported in Ryugu. The inferred glassy textures of SSO precursors in contrast are dissimilar to microchondrules reported from Ryugu [12,13], however, glassy microchondrules are common in chondritic meteorites owing to the more rapid cooling of these small droplets [17-20]."

We have also added EMPA analyses of the cores of microchondrules to the extended data to demonstrate these objects are chondritic.

Finally, I have updated the figure captions to describe objects as SSOs rather than microchondrules. None of the reviewers commented on this, however, it is bad practice to bias results.

Comment 2

There are some observations suggesting that the Ryugu original parent body have formed farther location from the Sun (farther than any other chondrite parent bodies and/or comet forming regions); carbonated water in a sulfide grain (Nakamura T. et al. 2022), $\delta^{15}\text{N}$ anomaly (Ito et al. 2022 Nat. Astron.), and Fe isotope anomaly along with Cr and Ti isotope anomalies (Hoppe et al. 2022 Sci. Adv.). Although the authors discussed about carbonated water in a sulfide grain, how about remaining two observations?

- The evidence from N and Fe-Cr-Ti isotope data for the highly primitive nature of Ryugu can be explained in the same way as the chemical composition. The fine-grained material is the least processed and preserved the primitive components. I have added both these papers to the penultimate paragraph as:

Line 288 " Previous studies of Ryugu have found higher $\delta^{18}\text{O}$, $\Delta^{17}\text{O}$, $\delta^{15}\text{N}$, $\epsilon^{54}\text{Cr}$ [9,11] than CI chondrites suggesting Ryugu is more primitive, and that the Fe, Ti and Cr isotope systematics

of Ryugu and CI chondrites are distinct from other carbonaceous chondrites, but similar to each other [46]. These observations have been interpreted to mean formation of Ryugu at greater heliocentric distance of formation, based on the assumption that processing of presolar materials, the carriers of nucleosynthetic signatures, and volatile materials increases sunwards in the early Solar System.” – this is followed by our existing suggestion that fine-grained materials, concentrated by turbulence, have experienced less processing.

Comment 3

(Not a major comment) Accretion location of the Ryugu original parent body should be explicitly indicated in Fig. 5.

- Added

Reviewer #2 (Remarks to the Author):

Comment 1

Currently, the manuscript, while clear and concise, does not intrude on the goals of the study or the scientific question or hypothesis to test.

I disagree, this is a rather rigid definition of the structure of a scientific paper, some do indeed have a goal and a clear hypothesis, others, like this one, are serendipitous discoveries. I think it is clear from the start what this paper is about. Nature Communications is full of excellent papers with similar structure ("we found this, isn't it significant") so there is precedent.

Comment 2

A discussion of the alteration process is needed for the non-specialist.

- I don't think a lengthy discussion is necessary. I have added a sentence on the alteration process for non-specialists:

Line 44 "Hydration occurred owing to the melting of ice by internal heating due to the decay of short-lived radio-isotopes. "

Comment 3

Are the supplementary materials missing.

I apologise, I realise that there were a couple of references to extended data, and none provided. These were ancillary, merely showing the shape factors. I have now added them. We have now had the opportunity to collect much more data and there is much more in the extended data than there was before. This should all add considerable weight to our identification of microchondrules.

Comment 4

My recommendation for the authors is to extract mineral information from the nano-CT, as done recently on Martian meteorites.

- This is unnecessary, we have characterised the composition of phases through elemental mapping of microchondrules in section. X-ray attenuation gives non-unique identifications in

particular for micron-sized grains where resolution and edge effects makes attenuation affected by the surrounding matrix. EDS analyses are far superior.

Comment 5

The study will benefit from a numeral model that supports Figure 5. It is unclear how this was constructed without a numerical way to test these hypotheses.

- Apologies, I mean't to specify what this figure is based upon. The figure caption now ends with:

"The figure is based on observations of a protoplanetary disk by Teague et al. [47]."

Reviewer 3

The study is compelling, and I look forward to seeing it published.

Comment 1

Turbulent concentration and size-sorting of fines of primitive carbonaceous materials may demand high levels of turbulence, higher than expected at 15-30 AU, rather than large heliocentric distance." It would be useful at this point of the discussion to indicate actual α numbers thought to be relevant for the inner and outer regions referenced.

I don't think the introduction is the best place for this kind of detail. Later in the discussion we do give alpha values. There is obviously no precise numbers to give, only estimates, but what I have added is:

Line 228 "whilst Desch et al [28] estimates values in the inner Solar System (<1 au) of $\alpha = 4 \times 10^{-3}$ with the outer Solar System (>10 AU) having $\alpha < 10^{-5}$ with α falling as a power law in between."

Later on in the discussion we already give values of alpha for the turbulent wake of Jupiter.

Comment 2

Perhaps they might consider writing instead "studies may have under-estimated their abundance."

Changed, quite right.

Comment 3

In lines 184-186 the authors write that "The smallest eddies in a turbulent system are defined by the Kolmogorov scale, at which kinetic energy is dissipated by viscosity into heat, and are those with the lowest vorticity." If the eddy overtime is being identified with the inverse vorticity (which is reasonable) then the opposite is true in Kolmogorov turbulence. The vorticity of an eddy of lengthscale ℓ in the inertial range of classic 3D isotropic turbulence goes like $\omega \ell \sim (L/\ell)^{2/3}$. Maybe the authors intended to say that the overturn times shorten with decreasing lengthscale?

Apologies, this is a typo, I of course mean the highest vorticity.

Comment 4

The authors write on lines 186-187 that "Particles with stopping times equivalent to the overturn time of the Kolmogorov scale eddy are preferentially concentrated by turbulence

[24,25]" The principle the authors are getting at are correct, but there have been several important advances in this picture since the publication of references [24, Leshin et al.] and [25, Cuzzi et al. 2006]. The key point is that preferential concentration does not necessarily occur at the Kolmogorov scale. Two recent studies, Hartlep et al. (2017) and Hartlep & Cuzzi (2020), demonstrate that for a given level of turbulence, α , and a specific particle size, d (with Epstein regime stopping time $t_s \equiv \rho_s(d/2) / \rho_g c_s$ where ρ_s is the solid particle density, ρ_g is the ambient gas density, c_s is the local sound speed), there exists a length scale within the inertial range of presumed 3D isotropic turbulence where enhanced clustering due to turbulent concentration is expected. The preferred length scale, denoted as ℓ_{clust} , is identified at the scale where the turbulent Stokes number, $St_\ell \equiv \omega \ell(\ell_{\text{clust}}) t_s$ is ≈ 0.3 .

This differs from the traditional expectation that the clustering scale should be the Kolmogorov scale, $\ell_\eta \sim Re^{-3/4} H$, where H is the local gas scale height and Re is the molecular Reynolds number. Equation (1), which the authors use to estimate a preferred d (derived from Cuzzi et al. 2001), is based on the assumption that the clustering scale is ℓ_η , thereby determining the corresponding particle size. In other words, the case where $\ell_{\text{clust}} = \ell_\eta$ represents a special limiting scenario in isotropic turbulence. Clustering does not have to happen on that scale.

Therefore, I am concerned that the inferences regarding the implied particle size and/or high levels of turbulence may be inaccurate if based solely on the application of Eq. (1): for any d there exists a lengthscale $\ell = \ell_{\text{clust}}$ in which clustering is maximal. The authors need to confront this possibility or state clear caveats as with regards to the use of Eq. (1).

- As a meteoriticist I am afraid I find Hartlep et al 2017 particularly inaccessible and difficult for a non-specialist to follow. The argument seems to be that direct numerical simulations of particle concentration in turbulence suggest particles of larger sizes are concentrated by turbulence than suggested by equation 1, specifically that equation 1 assumes concentration at a stokes number of 1, whilst Hartlep suggests 0.3. This result seems to be based on optimisation to produce scale invariance that is valid in the inertial range. It is unclear to me why this is necessarily the favoured solution. Whilst Desch et al. 2018 predicts the concentration of chondrule-size particles, which matches observation, Hartlep et al., 2017 predicts concentration of clusters of thousands of chondrules, meaning there is no process to concentrate chondrules, which is contrary to observation - although admittedly this is somewhat of a chicken and the egg argument. I would be much more comfortable if it were objections based on experimental data, rather than a choice between two models, both with inherent assumptions. This said, the reviewer has a valid point, there are two papers, by the same author casting doubt on the precision of Eqn 1. I acknowledge this issue in the paper by saying:

Line 218 "Equation 1 is based on the assumption that the clustering scale is the Kolmogorov scale and is thus a special limiting scenario in isotropic turbulence that leads to concentration of chondrule-sized objects in the inner Solar System. Direct numerical simulations of particle clustering in turbulence in contrast suggests clustering of larger particles in turbulence [29]."

Comment 5

In the discussion containing Lines 199-210, it is difficult to gauge the relative coupling of the particles to the gas. For the dynamicists it would be beneficial to write out an estimate for the corresponding Stokes number, $St \equiv t_s \Omega_K$, where Ω_K is the local Keplerian rotation frequency.

- This is somewhat problematic since it is not as simple as it sounds. The stokes number measures the coupling between gas and dust and thus depends on gas surface density at any heliocentric distance and is influenced by collisions and fragmentation (Batygin et al., 2022). Given these uncertainties I think it not useful to estimate a value subject to such uncertainty.

All this paper requires is the concept that with increasing turbulent intensity concentration of increasingly small particles occurs. We've relied on a simple analytical equation in order to dispense with these complexities. I hope the reviewer agrees this is more than most meteorites studies attempt. If there is a simple and rigorous way to specify the turbulent stokes number with heliocentric distance in the early solar system, without pages of exposition and caveats, then please let the reviewer educate me and I will be more than happy to include them.

Comment 6a

In the paragraph starting on line 238, the authors suggest that sunward of the inner edge, spiral wave shedding could trigger the Streaming Instability (SI). While this might be plausible, it is only likely if the turbulence is weak enough or if the particle Stokes numbers (see below) fall within the ideal range for SI to operate. In terms of turbulence, it has become increasingly evident in recent years that SI may not be triggered if the background turbulence is high and the Stokes numbers are low (below 0.01). The authors should consider adding a cautionary note in this regard by citing recent studies on the matter, such as Estrada & Umurhan (2023), Chen & Lin (2020), and Umurhan et al. (2020). More importantly, the authors should note that SI can indeed be triggered when $St = 0.1-1$ (see Li & Youdin, 2021 and Estrada & Umurhan, 2023), but this occurs only under the right turbulence conditions. Is the Stokes number of the $d = 7\mu\text{m}$ particle mentioned in line 203 within this preferred range? What is the precise value?

Again this is a very complex question. Estrada & Umurhan (2023) model suggests turbulence at α of 10^{-3} to 10^{-4} in the early solar system precludes streaming instability in the first few million years, contrary to the evidence for early formed planetesimals by meteorites. Of course another mechanism, such as pebble accretion, may be responsible for planetesimal growth, but these are subject to some of the barriers to growth that are circumvented by streaming instability. However, other papers such as Gerosa et al., (2023) show that radial shear in turbulent disks can favour anticyclones that concentrate particles into hierarchical clusters that merge with time. This doesn't seem to be considered by Estrada & Umurhan (2023) and might change their results. It seems the behaviour of particles in turbulent systems is very complex and small differences in assumptions can result in very different outcomes. As described above turbulent stokes numbers are somewhat problematic.

To acknowledge these issues I have added:

Line 277 "although recent models caution that turbulence can also resist the formation of dense enough regions to cause instability [43]."

Comment 6b

In lines 246-250, the authors also mention the possibility of very large turbulence, $\alpha \sim 10^{-2}$, triggering the SI. Specifically, I am referring to the statement in lines 249-250: "Turbulence is not only able to concentrate particles of the same size as microchondrules, it is likely to concentrate them to the levels needed to trigger streaming instability." This has not yet been demonstrated in any published numerical study of the SI (see the previously cited references). The authors should either revise this statement or include qualifying language to indicate that this remains an unverified hypothesis.

I have changed "likely" to "might" as suggested.

Response to Reviewers Comments

Abundant microchondrules in 162173 Ryugu suggests a turbulent alternative to the distant origins of primitive asteroids

In the following we respond to each of the reviewers comments and provide details of the changes made in italics.

Reviewer 1

The authors addressed my comments by modifying the main text and adding more images and EMPA data, so that one of my two concerns is cleared. But another one remains.

I recommend publication after addressing the concern and additional comments.

Comment 1

The argument that the Ryugu original parent body formed at ~3.2AU is not yet convincing.

An alternative scenario explaining the Ryugu isotope anomalies is quantitative and hard to imagine. Kawasaki et al. (2022) found that the ratios of ¹⁶O-rich to ¹⁶O-poor minerals in Ryugu and CI chondrites are higher than in other carbonaceous chondrites groups but are similar to that of comet 81P/Wild2 and suggested that Ryugu and CI chondrites accreted in the outer Solar System closer to the region of comets. How do you explain this observation in your scenario?

Response 1

This concern makes the assumption that the accreted parent bodies become increasingly primitive with distance from the sun. In our paper we propose that the concentration of fine-grained materials preferentially concentrates primitive materials that have escaped processing by virtue of their smaller grain size. This also clearly applies to ¹⁶O-rich minerals.

We already stated:

Line 311 "An alternative scenario for the formation of Ryugu is the inward cycling and concentration of fine-grained materials in or near the gap formed by Jupiter. The primitive nature of CI-like materials and the large abundance of presolar nuclides could be explained by their formation in regions that preferentially trap fine, largely unprocessed, primitive grains, and expel processed larger grains such as chondrules, whilst retaining microchondrules. Thus, CI chondrites might not be fine-grained because they are primitive, but they may be primitive because they are fine-grained."

To address the reviewers comment we have now mentioned Kawasaki et al (2022):

Line 301 "Furthermore, the ratio of ¹⁶O-rich/¹⁶O-poor minerals within Ryugu and CI chondrites is higher than other carbonaceous chondrites and similar to comet 81/P Wild-2 [48]."

Line 311 We have also added the words "The primitive chemical and isotopic nature of CI-like" to the paragraph on our alternative scenario.

Specific comments

L26 – Meridonal corrected to Meridional

- Corrected

L46 – objectsi corrected to objects

- Corrected

L46-47 Reference added to Weisberg et al 2006 Messi

- Added

L67 Commas added between ...) and 5.4-...anddiameter and were...

I wouldn't put comma's there...the copy editor can decide

L69 Period added between....Fig and 1-3...

Period added.

L78:What is 2-u?

Apologies it should read (Fig. 1a,u).

L85: What does CC in Fig. 1t stand for?

Figure caption 1. Definition now added: "(t) SSO C-44 exhibits a core with a faceted anhydrous silicate crystals set in a cryptocrystalline matrix (CC) consisting of phyllosilicate"

L133-136: Add a reference(s), if any

There are no references these are our alternative explanations. I think the lack of references makes this clear.

L190-191: Is it possible to describe the lithologies where the SSOs are observed. Since the SSOS are all replaced by phyllosilicate, the lithologies may not be the least- and less-altered.

Line 199. I have added "*Sample A0180 lacks anhydrous silicates and is more already than many Ryugu samples, however, it retains evidence for freeze-thaw processes and may have experienced low-temperature alteration [23].*" A more in-depth description of A0180 is given in the references.

L229: Add a period between...in between and This...

Added

L243: Analysis conditions of EMPA should be described.

Now added to the methods

Reviewer 2

No further suggestions

Reviewer 3

"Equation 1 is based on the assumption that the clustering scale is the Kolmogorov scale and is this a special limiting scenario in isotropic turbulence that leads to concentration of chondrule-sized objects in the inner Solar System. Direct numerical simulations of particle clustering in turbulence in contrast suggest clustering of larger particles in turbulence [29]."

This statement mischaracterizes the conclusions of Hartlep & Cuzzi (2020). Their work does not claim that larger-sized particles are necessarily the ones preferentially concentrated. The

core idea the Turbulent Concentration mechanism is that the strongest intermittency in particle accumulation occurs when the turbulent eddy Stokes number is approximately 0.3. This condition can arise under a broad range of local parameters, including turbulent intensity, length-scale of interest, local disk mass, particle size and other factors that collectively determine St_{teta} .

The point emphasized by Hartlep & Cuzzi (2020) is that while Desch et al (2020) assume a priori that maximal clustering occurs at the Kolmogorov scale, this is only a subset of the broader conditions that can yield $St_{\text{teta}} = 0.3$. Consequently, the revised text should clarify this point more accurately – and simply saying that HC20 predicts larger sized particles is incorrect.

Response 1

Steve Desch has suggested the following changes. Firstly we acknowledge that we base our model on Desch et al. (2018) rather than HC20 and then give a little more detail:

Line 208. “Particles with stopping times equivalent to the overturn time of the Kolmogorov scale eddy—i.e., those with Stokes numbers $St \sim 1$ relative to the Kolmogorov scale or $St \sim 10^{-4}$ relative to the largest-scale eddies—are preferentially concentrated by turbulence [24,25,26].”

Line 214 “Recently [29] argued for a different cascade model that predicts different sizes of particles concentrated in different sizes of eddies, and questioned whether particles with $St < 10^{-2}$ could be concentrated at all, due to disruption of clumps by ram pressure. Given that the size of chondrule concentrated in asteroids appears universally uniform, and given the success of disk models predicting the size of chondrules that would be concentrated [28], we will continue to use the findings of Cuzzi et al. [26].”

However, HC20 do state in their abstract “An important difference between these results and those of Cuzzi et al. is that particle growth by sticking must proceed to a radius range of at least one to a few centimeters for the IMF and meteoritical properties to be most plausibly satisfied. That is, as far as the inner nebula goes, the predominant “particles” must be aggregates of chondrules (or chondrule-size precursors) rather than individual chondrules themselves.” This statement would seem to be consistent with our characterisation of that paper. To address this issue we modified the statement to:

Line 228. “As mentioned above, direct numerical simulations of particle clustering in turbulence in contrast suggests clustering of larger particles in turbulence under the most likely conditions for planetesimal formation [29].”

Those interested should read HC20 and Desch et al 2018.

Comment 2

Please add turbulent stokes numbers

Response 2

We’ve now added the following for an estimate of the turbulent stokes number:

Line 278. “The sunward inner edge of the Jovian pressure bump is an intriguing possible environment for the concentration of small particles. At 3.2 AU and 4 Myr, the model of [28] predicts $\Sigma \approx 8 \text{ g cm}^{-2}$, $T = 90 \text{ K}$, which for $\alpha \approx 10^{-2}$ would predict $d = 5 \mu\text{m}$ using Equation 1. The Stokes numbers of such particles are $St = (2\pi)^{1/2} \rho_s a / \Sigma \approx 2 \times 10^{-4}$.”

Title: Abundant microchondrules in 162173 Ryugu suggests a turbulent alternative to the distant origins of primitive asteroids

Summary: In the present paper, sub-spherical objects (SSOs), i.e., microchondrules, are observed in the Ryugu sample A0180 using Nano X-ray CT and on the polished surface of the sample using SEM. The microchondrules (5.4 – 30.2 μm in size) consist of phyllosilicate and sulfide rims and are more abundant (350 ppm) than chondrule-like objects found in Nakashima et al. (2023 Nat. Commun.). The authors argue that the microchondrules have formed in the outer solar system chondrule factory beyond the Jupiter orbit at around 3 au and were transported inward near the Jupiter orbit, where the original parent body of Ryugu accreted. This means that the accretion location of the Ryugu original parent body is closer to the Sun than expected in the previous studies.

General comment: The present paper provides new insights into the evolution of solar system materials, especially the origin of microchondrules in solar system bodies beyond the Jupiter orbit and deserves to be published in Nature Communications. However, I have two concerns about the present paper, which are shown below. I recommend publication after addressing the concern and additional comments.

1. The argument that the SSOs are microchondrules is based only on their spherical shapes, which is weak. Since the SSOs and surrounding matrix are both composed of phyllosilicate, so that there is no mineralogical difference between them. In case of chondrule-like objects in the Ryugu samples (Nakashima et al. 2023), they consist of olivine and Fe-Ni metal and have ^{16}O -rich and -poor isotope ratios. Even if altered, it shows a barred olivine-like texture, so that it was recognized as a chondrule (Nakamura T. et al. 2022 Science).
If the authors observe less-altered clasts in the Ryugu samples, unaltered microchondrules like those (cryptocrystalline?) in ordinary and CM chondrites may be found, though such an object was not found in the clasts in the Ryugu samples as far as I have observed (Nakashima et al. 2023 Hayabusa Symposium).
2. There are some observations suggesting that the Ryugu original parent body have formed farther location from the Sun (farther than any other chondrite parent bodies and/or comet forming regions); carbonated water in a sulfide grain (Nakamura T. et al. 2022), $\delta^{15}\text{N}$ anomaly (Ito et al. 2022 Nat. Astron.), and Fe isotope anomaly along with Cr and Ti isotope anomalies (Hoppe et al. 2022 Sci. Adv.). Although the authors discussed about carbonated water in a sulfide grain, how about remaining two observations?
3. (Not a major comment) Accretion location of the Ryugu original parent body should be explicitly indicated in Fig. 5.

9/24/2024

Daisuke Nakashima

Review of “Abundant microchondrules in 162173 Ryugu suggests a turbulent alternative to the distant origins of primitive asteroids” by Genge et al., submitted to Nature Communications

October 2024

1 Summary Review of Study

The authors report the discovery of altered microchondrules, 6-8 μm in size, in the A0180 sample obtained from the C-type asteroid Ryugu. Their size and shape distributions mirror those of regular chondrules, suggesting similar formation processes. The authors go on to propose that these microchondrules formed in the outer Solar System near a Jovian pressure bump and were later transported sunward – all occurring under rigorous turbulent conditions. This finding challenges the notion that the most primitive asteroids necessarily formed at the farthest heliocentric distances.

2 Evaluation

I want to clarify that I am not an expert in the sample analysis techniques extensively discussed. I am operating under the assumption that these methods were executed properly and that the findings are accurate. My comments will focus solely on the nature of disk turbulence and the turbulence-related physical inferences presented by the authors.

The study is compelling, and I look forward to seeing it published. However, I believe there are significant issues with the turbulence estimates in their model, as well as with the broader discussion on turbulence in disks. I strongly encourage a revision that addresses these concerns, as doing so will only strengthen the manuscript and better convey the importance of their findings.

3 Points needing addressing

All of the following in no particular order of importance

1. Lines 61-63, the authors write, "Turbulent concentration and size-sorting of fines of primitive carbonaceous materials may demand high levels of turbulence, higher than expected at 15-30 AU, rather than large heliocentric distance." It would be useful at this point of the discussion to indicate actual α numbers thought to be relevant for the inner and outer regions referenced.
2. A minor wording matter, the authors write in lines 172-174 that "However, only five microchondrules were observed on the polished surface in sample A0180,

with the majority detected by Nano-XCT, thus previous studies have likely under-estimated their abundance." The tone suggested here is that the authors are certain that these are underestimates, but they haven't demonstrated that this is the case either. Perhaps they might consider writing instead "studies may have underestimated their abundance."

3. In lines 184-186 the authors write that "The smallest eddies in a turbulent system are defined by the Kolmogorov scale, at which kinetic energy is dissipated by viscosity into heat, and are those with the lowest vorticity." If the eddy overturn time is being identified with the inverse vorticity (which is reasonable) then the opposite is true in Kolmogorov turbulence. The vorticity of an eddy of lengthscale ℓ in the inertial range of classic 3D isotropic turbulence goes like $\omega_\ell \sim (L/\ell)^{2/3}$. Maybe the authors intended to say that the overturn times shorten with decreasing lengthscale?
4. The authors write on lines 186-187 that "Particles with stopping times equivalent to the overturn time of the Kolmogorov scale eddy are preferentially concentrated by turbulence [24,25]" The principle the authors are getting at are correct, but there have been several important advances in this picture since the publication of references [24, Leshin et al.] and [25, Cuzzi et al. 2006].

The key point is that preferential concentration does not necessarily occur at the Kolmogorov scale. Two recent studies, Hartlep et al. (2017) and Hartlep & Cuzzi (2020), demonstrate that for a given level of turbulence, α , and a specific particle size, d (with Epstein regime stopping time $t_s \equiv \rho_s(d/2)/\rho_g c_s$ where ρ_s is the solid particle density, ρ_g is the ambient gas density, c_s is the local sound speed), there exists a length scale within the inertial range of presumed 3D isotropic turbulence where enhanced clustering due to turbulent concentration is expected. The preferred length scale, denoted as ℓ_{clust} , is identified at the scale where the turbulent Stokes number, $St_\ell \equiv \omega_\ell(\ell_{clust})t_s$ is ≈ 0.3 .

This differs from the traditional expectation that the clustering scale should be the Kolmogorov scale, $\ell_\eta \sim Re^{-3/4}H$, where H is the local gas scale height and Re is the molecular Reynolds number. Equation (1), which the authors use to estimate a preferred d (derived from Cuzzi et al. 2001), is based on the assumption that the clustering scale is ℓ_η , thereby determining the corresponding particle size. In other words, the case where $\ell_{clust} = \ell_\eta$ represents a special limiting scenario in isotropic turbulence. **Clustering does not have to happen on that scale.**

Therefore, I am concerned that the inferences regarding the implied particle size and/or high levels of turbulence may be inaccurate if based solely on the application of Eq. (1): for any d there exists a lengthscale $\ell = \ell_{clust}$ in which clustering is maximal. The authors need to confront this possibility or state clear caveats as with regards to the use of Eq. (1).

5. In the discussion containing Lines 199-210, it is difficult to gauge the relative coupling of the particles to the gas. For the dynamicists it would be beneficial to write out an estimate for the corresponding Stokes number, $St \equiv t_s \Omega_K$, where Ω_K is the local Keplerian rotation frequency.
6. In the paragraph starting on line 238, the authors suggest that sunward of the inner edge, spiral wave shedding could trigger the Streaming Instability (SI). While this might be plausible, it is only likely if the turbulence is weak enough

or if the particle Stokes numbers (see below) fall within the ideal range for SI to operate. In terms of turbulence, it has become increasingly evident in recent years that SI may not be triggered if the background turbulence is high and the Stokes numbers are low (below 0.01). The authors should consider adding a cautionary note in this regard by citing recent studies on the matter, such as Estrada & Umurhan (2023), Chen & Lin (2020), and Umurhan et al. (2020).

More importantly, the authors should note that SI can indeed be triggered when $St = 0.1 - 1$ (see Li & Youdin, 2021 and Estrada & Umurhan, 2023), but this occurs only under the right turbulence conditions. Is the Stokes number of the $d = 7\mu\text{m}$ particle mentioned in line 203 within this preferred range? What is the precise value?

In lines 246-250, the authors also mention the possibility of very large turbulence, $\alpha \sim 10^{-2}$, triggering the SI. Specifically, I am referring to the statement in lines 249-250: "Turbulence is not only able to concentrate particles of the same size as microchondrules, it is likely to concentrate them to the levels needed to trigger streaming instability." This has not yet been demonstrated in any published numerical study of the SI (see the previously cited references). The authors should either revise this statement or include qualifying language to indicate that this remains an unverified hypothesis.